# Evaluation of 'Lorca' Cultivar Aptitude for Minimally Processed Artichoke

**Marina Giménez-Berenguer** [1], **María E. García-Pastor** [1], **Santiago García-Martínez** [2], **María J. Giménez** [1,*] and **Pedro J. Zapata** [1]

1   Department of Food Technology, EPSO, University Miguel Hernández, Ctra. Beniel km. 3.2, 03312 Alicante, Spain; marina.gimenez02@goumh.umh.es (M.G.-B.); m.garciap@umh.es (M.E.G.-P.); pedrojzapata@umh.es (P.J.Z.)
2   Centro de Investigación e Innovación Agroalimentaria y Agroambiental (CIAGRO-UMH), University Miguel Hernández, Ctra. Beniel km. 3.2, 03312 Orihuela, Spain; sgarcia@umh.es
*   Correspondence: maria.gimenezt@umh.es; Tel.: + 34-966-749-798

**Abstract:** Previous research works have reported that 'Lorca' artichoke cultivar presents a lower total phenolic content than other cultivars rich in phenolic compounds, which could show a lower susceptibility to enzymatic browning and increase its aptitude for fresh-cut processing. The aim of this study was to analyze the total phenolic content as well as browning evaluation by image analysis and polyphenol oxidase (PPO) enzyme activity in 'Lorca' cultivar in order to characterize the key factors which influence its phenolic levels for minimally processed artichokes. Thus, artichokes were harvested and classified on three head orders (main, secondary, and tertiary), as well as three development stages (initial, intermediate, and advanced). Variance components analysis was carried out for total phenolic content considering three factors: plant, flower head order, and internal development stage. For the first time, the internal development stage has been related to total phenolic content, and results showed that artichoke head order and internal development stage were responsible for a variability of 22.17% and 15.55%, respectively. Main artichoke heads and those at the advanced development stage presented the lowest phenolic concentration as well as the lowest PPO activity; therefore, they exhibit the lowest browning process, which could increase their use in ready-to-eat products at market.

**Keywords:** total phenol content; browning susceptibility; artichoke head order; internal development stage; postharvest

## 1. Introduction

Globe artichoke, *Cynara cardunculus* L. var. *scolymus* (L.) Fiori, is an herbaceous perennial plant belonging to the family Asteraceae, and it is native from the Mediterranean region. Currently, its production and consumption have spread worldwide since it is a product highly appreciated by consumers, not only due to its flavor, but also because its intake provides benefits for human health and well-being [1]. Nowadays, changes in the society lifestyle have led to an increased demand of minimally processed products considered as added value products in terms of quality, convenience, nutritional value, and ease of preparation [2]. In this sense, processed artichokes as fresh-cut products would provide a good solution to increase their consumption due to the high percentage of discarded plant waste, complexity of preparation, and trimming operations. However, cutting operations increase respiration rate and accelerate senescence and enzymatic browning, leading to a reduction of shelf-life [3].

The nutritional importance of artichoke heads is predominantly related to its high polyphenol content [4,5], mainly hydroxycinnamic acids and flavonoids, such as luteolins [6,7]. However, quantitative and qualitative phenolic profile is influenced by different preharvest factors: genotype, harvest date, environmental conditions, and agronomic

management, among others [5,6,8]. Different authors have reported that environmental conditions influence phenol biosynthesis for artichoke genotypes and parts of the plant, as well as there being a negative correlation between phenolic content and the age of plant tissues [1,9,10]. Significant seasonal fluctuations in both phenolic content and profile were observed, which have been attributed to a combination of ambient air temperature and solar radiation [11,12]. Similarly, Lombardo et al. [13] observed that phenol concentration was higher in artichokes harvested in the spring than winter-harvested artichoke heads. Contrary, other authors observed that artichoke heads harvested in the winter season showed a better quality than artichokes harvested in the spring [14,15]. Nevertheless, minimally processed artichoke heads showed higher susceptibility to browning in the spring season than in winter, even if the polyphenol content of artichokes harvested in winter was higher compared to spring buds [16].

Phenolic compounds are highly susceptible to browning through oxidation reaction catalyzed by polyphenols oxidase (PPO) and peroxidase (POD) enzymes and the *o*-quinones formed could take part in a second non-enzymatic step, leading to the formation of dark compounds [3,17]. Enzymatic browning is influenced by many variables which are dependent on the genotype, such as the specific activity of enzymes or the quantity and nature of phenolic compounds [18]. Therefore, cultivars rich in phenolic compounds and with a high enzyme activity are unsuitable for fresh-cut processing [3,19,20]. Several studies have showed that genetic material is a key determinant of phenolics content, antioxidant enzyme activities, and thus, the suitability of artichoke cultivar for fresh-cut processing, which contributes to the overall functional quality of this crop [10,21,22]. Phenolic content was significantly lower in seed-propagated cultivars than in the vegetatively propagated ones [23]. In this sense, cultivars with a low content of total phenols could produce artichoke heads with the best quality traits for fresh-cut and agro-processing industry, reducing the susceptibility to browning and the use of antioxidant treatments in processing steps [23].

On the other hand, one study recently published has reported that individual phenolic compounds, mainly hydroxycinnamic acids and luteolin derivatives, were highly influenced by flower head order (main, secondary, and tertiary heads) in seed-propagated open-pollinated, vegetatively propagated, and seed-propagated hybrid artichoke cultivars [7]. In this work, the individual phenolic profile was similar in all cultivars studied, reporting that 5-*O*-caffeoylquinic acid was the main hydroxycinnamic acid, followed by 3,5-di-*O*-caffeoylquinic acid and 3,4-di-*O*-caffeoylquinic acid [7]. Regarding the flower head order, tertiary head orders showed the highest individual phenolic content, followed by secondary and main heads, and this effect was cultivar dependent. Among eight different artichoke cultivars studied, 'Lorca' cultivar presented a lower total phenolic content than other cultivars rich in phenolic compounds, which could increase its aptitude for fresh-cut processing [7]. Thus, the main aim of this study was to evaluate the influence of flower head order and internal development stage on phenol content as well as browning susceptibility in 'Lorca' cultivar, in order to characterize the key factors which influence its phenolic levels and determine its ready-to-eat artichoke aptitude.

## 2. Materials and Methods

### 2.1. Plant Material and Experimental Design

'Lorca' cultivar was studied throughout the developmental cycle of one season (from August 2020 to April 2021), in an experimental plot of the Miguel Hernández University (Orihuela, Southeast Spain; 38°06′63.52″ N, −0°988′09.29″ W). The growing area has a Mediterranean climate, characterized by mild and humid winters and hot and rainless summers. Maximum monthly temperature ranges between 16.2 °C (January) and 33.3 °C (July) and minimum temperature between 5 °C (January) and 21.2 °C (August) were recorded, according to historical meteorological data.

For the experiment, 40 artichoke plants of 'Lorca' cultivar, which were reproduced by seeds and open pollination, were planted in mid-August following a planting frame of 0.8 m apart within a row and 1.2 m apart among close rows. These plants were distributed

in four replicates or blocks (10 plants planted per replicate or block). Crop management was performed according to standard agronomic practices used by growers in Southeast Spain. An application of commercial sheep manure was incorporated before transplanting, at a rate of 2.5 kgm$^{-2}$. Fungicides and insecticides were used along the developmental cycle, and fertilizers, composed of 250 kg N, 120 kg P$_2$O$_5$, and 300 kg K$_2$O per ha were applied by drip irrigation system. Gibberellic acid was not applied.

All artichokes of each plant were harvested following commercial criteria, when flower heads reached their average commercial size and morphology, and they were firm and tightly closed. The first harvest date was performed on February 1st, and artichoke heads were harvested weekly. A total of 13 different harvest dates along the complete crop cycle were performed, the last one being on 21 April. For each harvest date, artichokes were classified according to the flower head order: main, secondary, and tertiary heads, and a total of 445 artichokes were harvested (Table 1).

**Table 1.** Number of artichokes produced in each type of flower head order for 'Lorca' cultivar.

| Flower Head Order | Number of Artichokes |
|---|---|
| Main | 40 |
| Secondary | 278 |
| Tertiary | 127 |
| Total | 445 |

After harvest, artichokes were transported to the laboratory and analyzed individually on the same day. An evaluation of the internal development stage was performed using a three-level scale (Figure 1): 1 = initial development stage (artichokes without internal hair or with a small amount of short hair as well as with a green and tender hearts); 2 = intermediate development stage (artichokes with an intermediate amount of internal hair, but still showing a green heart); 3 = advanced development stage (artichokes with a high amount of internal hair, although appreciating a change from green to purple color on the artichoke heart). For PPO analysis, a half of the artichoke heart was quickly sliced, frozen with liquid nitrogen, and stored at −80 °C.

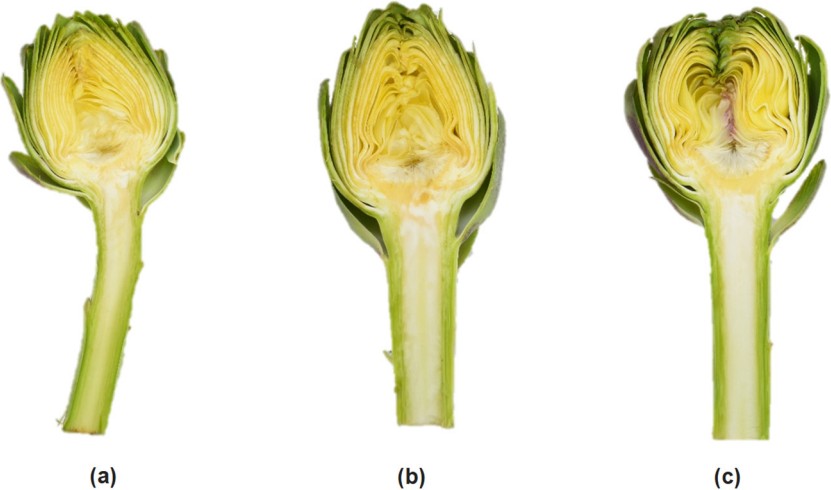

**(a)**　　　　　　　　　**(b)**　　　　　　　　　**(c)**

**Figure 1.** Scale of internal development stage: (**a**) initial development stage; (**b**) intermediate development stage; (**c**) advanced development stage.

## 2.2. Extraction and Quantification of Total Phenolic Compounds

Phenolic extraction was performed according to Martínez-Esplá et al. [24]. Briefly, 5 g of the edible part of each artichoke (heart and inner bracts) was homogenized with 15 mL of methanol at 80% containing 2 mM sodium fluoride (NaF), in order to inactivate polyphenol

oxidase activity and prevent phenolic degradation, for 2 min using a homogenizer (Ultra-Turrax®, TP 18, IKA, Staufen, Germany). Subsequently, samples were centrifuged at 10,000 g for 15 min at 4 °C. The supernatant was used for total phenolic quantification in duplicate using the Folin-Ciocalteu reagent and results (mean ± SE) were expressed as grams of gallic acid equivalent per $kg^{-1}$ of fresh weight (FW).

*2.3. Browning Evaluation by Digital Image Analysis*

Color was expressed by hue angle and measured with digital image analysis. Images of artichoke heads were captured using a digital camera (Nikon D3400, Minato, Tokio, Japan) in a light box with a white background. The camera setup conditions were as follows: light provided by two LEDs with a color temperature of 5600 K, speed of 1/5 s, ISO−100, focal aperture (f) 20, and length of 35 mm. For each artichoke head, a cut in half was made individually and longitudinally. An image was captured every 15 s for 3 min of the internal front side to evaluate the browning process after cutting each individual artichoke. Images were saved as JPEG files and analyzed using ImageJ v1.52a software (NIH Image, National Institutes of Health, Bethesda, USA). The CIELab model was used to calculate the hue angle (h°) with the color parameters a* and b*, as follows (1):

$$\text{hue angle} = \arctan (b^{*}/a^{*}) \tag{1}$$

*2.4. PPO Enzyme Activity Analysis*

The extraction and quantification of PPO enzyme activity was carried out according to Cabezas-Serrano et al. [3] with slight modifications. Briefly, 5 g of frozen tissue were homogenized (Ultra-Turrax®, TP 18, IKA, Staufen, Germany) in 15 mL of 0.1 M phosphate buffer (pH 6.5) containing 1 g of polyvinylpyrrolidone (PVP) and centrifuged at $10,000\times g$ for 15 min at 4 °C. The supernatant was used for PPO activity determination which was spectrophotometrically performed using catechol as a phenolic substrate. The reaction mixture contained 200 μL of extract, 2 mL of 50 mM phosphate buffer and 0.5 mL of 0.1 M 4-methylcatechol and measured by determining the absorbance increase at 420 nm. The results were expressed as units of enzyme activity per g of FW, and one enzyme unit was defined as the amount of enzyme causing an increase in absorbance of 0.001 per min.

*2.5. Statistical Analysis*

An analysis of the variance components was carried out with all the phenol content values in order to estimate the amount of variability provided by each of the three factors studied in the experiment: plant, flower head order, and internal development stage. The data for the analytical determinations were subjected to analysis of variance (ANOVA), the dependent variable being the total phenolic content and the three factors indicated above. Mean comparisons were made using a multiple range test (Newman–Keuls test) to determine significant differences at $p < 0.05$ among artichoke head orders or among internal development stage in each parameter. Significant differences on Δ hue angle were performed according to *t* Student test at $p < 0.05$ among artichoke head orders. A correlation analysis was performed between phenolic content and PPO activity or hue angle increment. All analyses were performed with the STATGRAPHICS Plus software package, version 3.1 for Windows.

**3. Results and Discussion**

*3.1. Influence of Variance Components Analysis on Phenolic Content*

Table 2 shows the percentage (%) of each variance component studied after the statistical analysis for the phenolic content, showing the importance attributed to each factor. The analysis of variance divides the variance of phenolic content (dependent variable) into three components, one for each factor (Table 2). Each factor after the first one is nested in the one above. With this analysis, we estimate the amount of variability contributed by

each of the studied factors. The three factors analyzed in this study, in addition to total phenolic content, were: plant, artichoke head order, and the internal development stage.

**Table 2.** Analysis of variance components for phenolic content for 'Lorca' cultivar.

| Source | Percentage (%) |
| --- | --- |
| Plant | 4.55 |
| Artichoke head order | 22.17 |
| Internal development stage | 15.55 |
| Error | 57.74 |

Our results reported for the first time that the three variance components analyzed (plant, artichoke head order, and internal development stage) explained a 42.27% of all variances. The plant only showed a variability of 4.55%; therefore, this variance component did not have an important role on the total phenolic content since all plants studied presented similar results and only one plant of the 40 plants in total showed significant differences. This result could be related to the fact that the plants are propagated by seeds, which have been selected over time and present a high homogeneity. According to our results, the most important factors were artichoke head order and internal development stage, which presented a variability of 22.17% and 15.55%, respectively. The remaining variability of 57.74%, namely as error, could be explained by different preharvest factors. In this sense, quantitative and qualitative variability of essential nutrients and secondary metabolites in artichoke heads could depend on genetic diversity, environmental conditions, edaphic, harvest time, and climatic conditions throughout the artichoke plant growth cycle, as it was previously reported by other authors [8,11,13,25].

All artichokes were harvested in 3 months, from February to April; however, harvest time was not considered for the variance components analysis mainly due to no significant differences being found previously between winter and spring harvest season for 'Blanca de Tudela' cultivar [26]. Specifically, in 'Lorca' cultivar, the influence of gibberellic acid (GA$_3$) treatment on the phenolic content was evaluated, and no significant differences were observed with respect to untreated artichokes, which were harvested later [7]. On the other hand, other authors have observed that harvest date significantly influenced the total phenol content during longer developmental cycles. Specifically, Pandino et al. [11] evaluated the effect of harvest time in 'Ovoli' artichoke from November to April and reported higher phenolics concentration in artichoke heads harvested from February to April compared to previous ones. Thus, the higher phenolic level observed by other authors in those artichokes harvested in spring compared to winter season could be related to the fact that in spring, tertiary artichoke heads are mainly harvested and therefore, they have the highest phenolic content.

*3.2. Influence of Artichoke Head Order and Internal Development Stage on Total Phenolic Content*

The influence of artichoke head order on total phenolic content was evaluated to corroborate previous results obtained in the 2018–2019 growing season [7]. Eight white artichokes genotypes were studied, and total phenolic content showed significant differences among flower head orders and the effect was also cultivar dependent. As a general trend, all cultivars analyzed had a higher total phenolic content in tertiary flower head orders than in secondary or main ones. Specifically, 'Lorca' cultivar showed significant differences between main and tertiary heads but not between main and secondary artichokes [7]. Results of this 2020–2021 growing season showed that total phenolic content in 'Lorca' cultivar significantly varied among flower head orders (main, secondary, and tertiary). Accordingly, tertiary flower heads had significantly higher levels of phenolic compounds than secondary and main heads, the main ones being those artichokes with the lowest content (Table 3). The greater significant differences observed in the present study among the three flower head orders could be related to the higher plant number included in the present experimental design as well as the number of artichokes analyzed, which was also higher

since all flower heads produced by each plant were analyzed individually in this study. Similarly, Gagliardi et al. [27] observed that the content of both 5-*O*-caffeoylquinic acid and 1,5-*O*-dicaffeoylquinic acid were approximately 2-fold higher in processing or tertiary heads class compared to main heads class in 'Violeto di Provenza' artichoke cultivar.

**Table 3.** Total phenolic content (g kg$^{-1}$) for different artichoke head orders (main, secondary, and tertiary heads) of 'Lorca' cultivar.

| Artichoke Head Order | Total Phenolic Content |
|---|---|
| Main heads | 2.155 ± 0.065 c |
| Secondary heads | 2.782 ± 0.023 b |
| Tertiary heads | 3.221 ± 0.019 a |

Data are the mean ± SE. Different lowercase letters show significant differences on total phenolic content among artichoke head orders, according to Newman-Keuls test at $p < 0.05$.

On the other hand, the internal development stage of artichoke heads for each harvest date has been evaluated and correlated to total phenolic content. Our results demonstrated for the first time that total phenolic content was significantly higher when 'Lorca' artichokes were in an initial (stage 1) or intermediate (stage 2) development stage. Contrary, the concentration of these bioactive compounds was lower at the most advanced development stage (stage 3) (Table 4). In the same way, Lattanzio et al. [28] observed that phenolics levels in artichoke heads depended on the stage of physiological organ, being higher in younger tissues. In addition, it has been observed that the development stage or the age of tissue influenced the PPO activity, found the highest activity values in the heart, which represents the younger tissue of artichoke heads [28].

**Table 4.** Total phenolic content (g kg$^{-1}$) for different internal development stages (1: initial; 2: intermediate; 3: advanced) of 'Lorca' cultivar.

| Internal Development Stage | Total Phenolic Content |
|---|---|
| 1 (initial) | 2.889 ± 0.036 a |
| 2 (intermediate) | 2.827 ± 0.032 a |
| 3 (advanced) | 2.411 ± 0.023 b |

Data are the mean ± SE. Different lowercase letters show significant differences on total phenolic content among internal development stage, according to Newman–Keuls test at $p < 0.05$.

Due to the high proportion of artichoke plant waste and the complexity of preparation and trimming operations, processing artichokes as a fresh-cut product will provide convenience. Furthermore, since receptacle and inner bracts contain high amounts of phenols, fresh-cut artichokes would also provide a high value-added product. However, the main problem of fresh-cut artichokes is the cell membranes rupture associated to cutting processes, which induces enzymatic browning from the oxidation of phenolic compounds by PPO and POD enzymes and it results in the formation of dark compounds, leading to a high browning rate of the cut surface (artichoke receptacle and bracts) [3,29].

Artichokes with high levels of total phenols can be unsuitable for ready-to-eat products, since they are associated with either enzymatically or non-enzymatically oxidative browning reactions, mainly during processing steps [19,30]. As a consequence, main artichokes and artichokes at an advanced internal development stage could be particularly suited to fresh-cut processing, since they showed the lowest total phenolic content observed. Contradictorily, artichokes with a high total phenolic level are probably the best suited for fresh consumption, since the presence of these compounds in the human diet is correlated to a protective effect against certain chronic and degenerative diseases related to oxidative stresses [8,31,32]. In this sense, tertiary and secondary artichoke heads and artichokes at initial and intermediate development stages had the highest total phenolic content and, hence, they also show potential use for fresh consumption on healthy diets [19,33].

### 3.3. PPO Activity for Different Artichoke Head Orders

PPO activity was analyzed in order to complete the evaluation of the artichoke orders which are the most susceptible to browning after cutting (Table 5). The results showed significant differences between main and secondary or tertiary heads; however, no significant differences were found between secondary and tertiary heads. Similar levels for PPO activity have been reported in different artichoke cultivars at harvest, with values from 400 U g$^{-1}$ FW in 'Harmony' and 'Opal' up to 600 U g$^{-1}$ FW in 'Concerto' and 'Symphony' cultivars. Nevertheless, PPO activity was variable during the storage period, this trend being observed cultivar dependent [8].

**Table 5.** Polyphenol oxidase activity (U min$^{-1}$ g$^{-1}$) for different artichoke head orders (main, secondary, and tertiary heads) of 'Lorca' cultivar.

| Artichoke Head Order | PPO Activity (U g$^{-1}$ FW) |
| --- | --- |
| Main heads | 311.30 ± 19.05 b |
| Secondary heads | 411.90 ± 19.71 a |
| Tertiary heads | 451.31 ± 20.17 a |

Data are the mean ± SE. Different lowercase letters show significant differences on polyphenol oxidase activity among artichoke head orders, according to Newman-Keuls test at *p* < 0.05.

This is the first time that PPO enzyme activity has been analyzed according to the artichoke head order classification: main, secondary, and tertiary, in 'Lorca' cultivar. The correlation analysis between phenolic content and PPO activity was carried out and significant correlations were found for main ($R^2$ = 0.732), secondary ($R^2$ = 0.821), and tertiary ($R^2$ = 0.854) artichoke heads. These results agree with Cabezas-Serrano et al. [34], who found a significant coefficient of correlation between phenolic content and PPO activity, highlighting that PPO activity increased with the increment of phenolic content. The lowest phenolic content and PPO activity in main artichokes could be related with a lower increment of hue angle.

### 3.4. Evaluation of Artichoke Browning for Different Artichoke Head Orders

Color can be measured objectively and quickly by computerized image analysis techniques. The RGB (red, green, and blue) linear color model was used to identify the browned pixels. Starting from the RGB image by using a software that can convert the original image to non-linear coordinates L*, a*, b* (lightness, redness, and yellowness) [35]. The most critical step in image processing is to isolate the region to be studied (which can be the complete image or only a part) within the image, and this is done through an image segmentation [36].

Flower head order was the variance component with the greatest influence on phenolic content; therefore, an image analysis was carried out individually according to artichoke classification in main, secondary, and tertiary head orders. To evaluate color changes related to browning intensity of cut surface in the different artichoke orders analyzed, the increment of hue angle, such as the difference between hue angle at 180 s and at time 0, was obtained [37,38] by using ImageJ software for digital image analysis.

Total phenol content was correlated with the increase on hue angle variation. In general, artichokes underwent color changes after cutting, from yellow green to dark brown, which explained the decrease on hue angle over time [3,34]. Therefore, the increase on the hue color parameter, measured during 180 s, was higher for tertiary artichokes followed by secondary and main heads (Figure 2). For each artichoke head order, the results obtained were fitted to a third polynomial regression and a high correlation has been found between the increment of hue angle and time ($R^2$ = 0.9924 − 0.9975). These differences are shown by an increase on browning process in the fresh product after cutting up to 180 s for each different flower head order, with tertiary heads being those artichokes that presented the higher color changes due to their greater phenolic content (Figure 3). These results are according to those reported previously by other authors, where a high

correlation of browning score and hue angle on cut iceberg lettuce was observed [39,40]. Moreover, phenolic concentration could be directly correlated to browning susceptibility in fresh-cut artichokes since its initial high phenolic content may induce the browning reaction [41]. During the industrial processing of these vegetables, a greater impact of enzymatic browning is expected, consequently derived from mechanically damaged tissues. The browning process may be attributed to the interaction between phenols and iron due to an oxidation of iron ions and the subsequent formation of quinones catalyzed by PPO enzymes, supported by a reduction on the phenolic content [3].

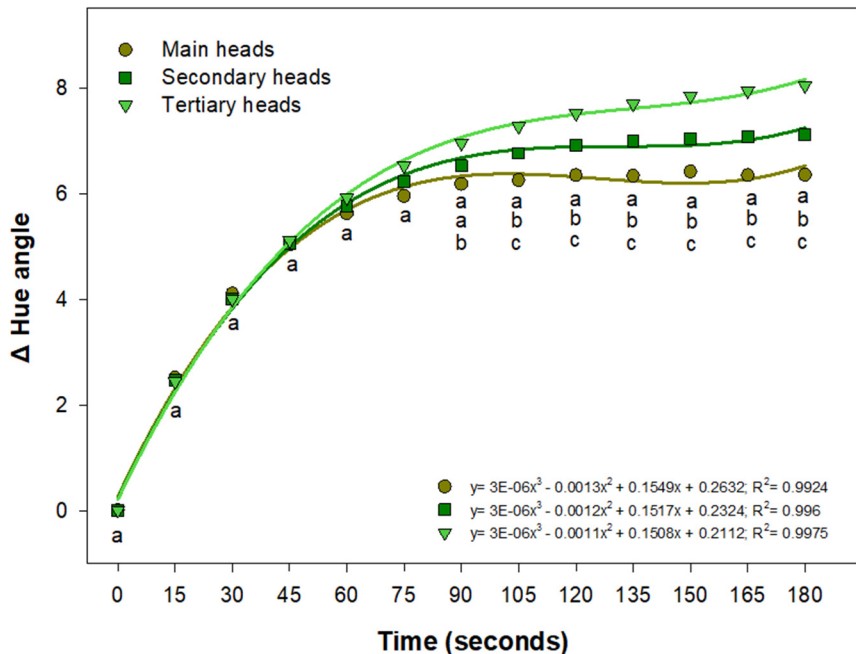

**Figure 2.** Effect of colour change (Δ Hue angle) of fresh-cut artichoke heads every 15 s for 3 min. Different letters show statistical differences (*t* Student test, *p* < 0.05) among artichoke head orders for 'Lorca' cultivar.

The results obtained in the present study in 'Lorca' artichoke cultivar showed differences on the susceptibility to browning attending to the flower head order. Tertiary and secondary heads showed the highest color changes and browning since they presented the highest phenolic content. On the contrary, main heads, which showed the lowest phenolic content, were the least susceptible to post-cutting browning [42]. A correlation analysis was performed between phenolic content and the increment of hue angle. Both parameters showed significant coefficients of correlation in main ($R^2$ = 0.775), secondary ($R^2$ = 0.891), and tertiary ($R^2$ = 0.856) artichoke heads. In previous works, it was observed that phenolic content was correlated with the susceptibility to browning in plant species rich in these compounds, such as artichokes, with these color changes being related to PPO enzyme activity [41]. On the other hand, browning was correlated with phenylalanine ammonia lyase (PAL) enzyme activity in species with a low phenolic content, such as lettuce [43].

In summary, our results highlight that the lowest phenolic content and PPO activity are related to the lowest increment of hue angle in main artichoke order, with the opposite effect being observed in secondary and tertiary. In general, tertiary heads showed the worst aptitude to developed fresh-cut artichoke product related to a high browning on the artichoke head (Figure 3). PPO activity is the key enzyme involved in the enzymatic browning process, which generates dark pigments called melanoidins, and are unacceptable in terms of safety as they could support microbial growth [44]. Several studies have showed that the browning process might be prevented by chemical and physical methods to inhibit PPO enzyme activity, including the reduction of temperature and/or oxygen concentration, the use of modified atmosphere packaging (MAP), and the application of anti-browning

agents, coatings, and innovative packaging [3,29,45–48]. The different methods used as PPO inhibitors could inhibit the PPO enzyme, removing its substrates or lowering pH below the optimum value for PPO [49]. For the first time, these results could be used as a tool to classify the artichoke head orders in 'Lorca' cultivar according to its ready-to-eat aptitude.

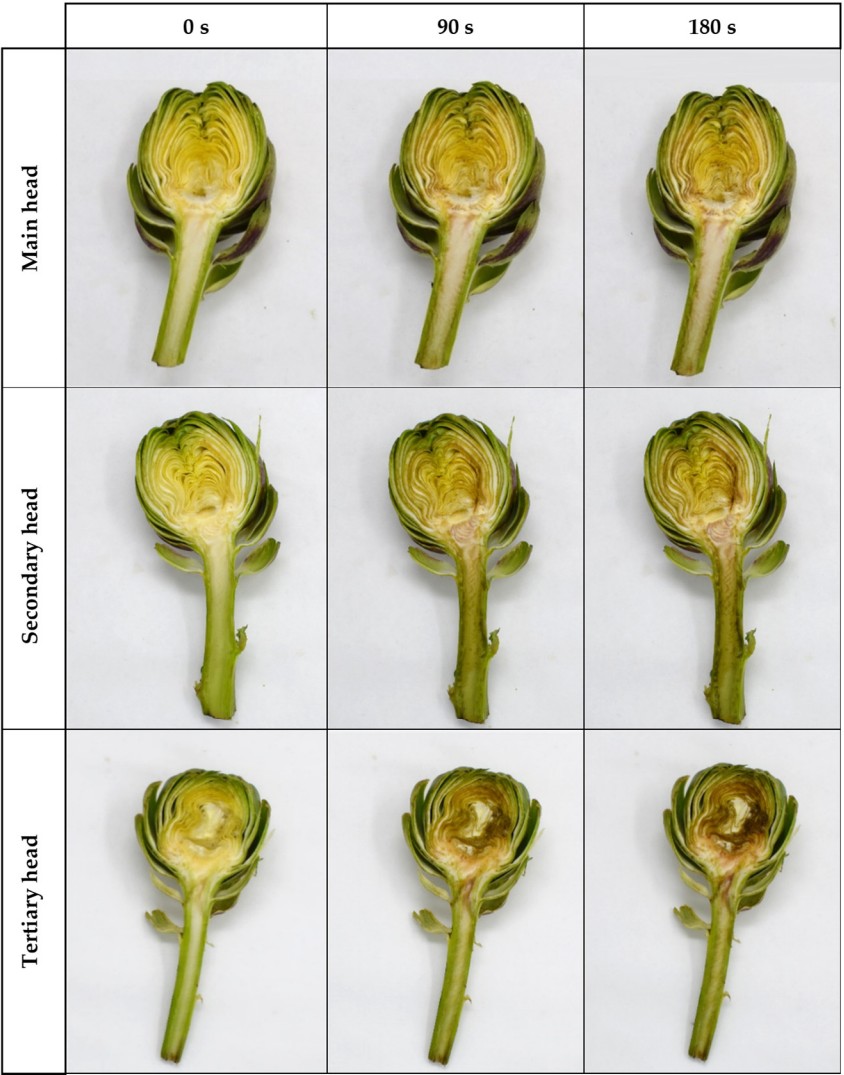

**Figure 3.** Enzymatic browning process of different flower head orders at time 0 s, 90 s, and 180 s for 'Lorca' cultivar.

## 4. Conclusions

In this study, a characterization of the total phenolic content in 'Lorca' cultivar has been performed to define their optimum commercial aptitude to minimally processed artichokes. Variance components analyses were carried out for total phenolic content considering three factors: plant, flower head order, and internal development stage. Results showed that the plant only explained a 4.55% of variability, with artichoke head order and internal development stage responsible for a variability of 22.17% and 15.55%, respectively. For the first time, the internal development stage has been related to total phenolic content. Tertiary and secondary artichokes and those at initial and intermediate development stages showed the highest phenolic content and PPO enzyme activity, showing the highest browning susceptibility after cutting. On the contrary, the opposite effect was observed in main artichokes and those at the advanced development stage; therefore, they exhibit the lowest browning process, which could increase their use in ready-to-eat products at market.

**Author Contributions:** Conceptualization, M.G.-B., M.J.G. and P.J.Z.; methodology, M.G.-B. and M.E.G.-P.; software, M.G.-B. and S.G.-M.; investigation validation, M.J.G. and P.J.Z.; formal analysis, M.E.G.-P.; investigation, M.G.-B.; resources, P.J.Z.; data curation, M.J.G.; writing—original draft preparation, M.G.-B.; writing—review and editing, M.G.-B., S.G.-M., M.E.G.-P., M.J.G. and P.J.Z.; visualization, M.G.-B.; supervision, M.J.G. and P.J.Z.; project administration, P.J.Z. All authors have read and agreed to the published version of the manuscript.

**Funding:** This research received no external funding.

**Institutional Review Board Statement:** Not applicable.

**Informed Consent Statement:** Not applicable.

**Data Availability Statement:** Not applicable.

**Conflicts of Interest:** The authors declare no conflict of interest.

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
