# Peer review of "Evaluation of ‘Lorca’ Cultivar Aptitude for Minimally Processed Artichoke"

_agronomy, doi:10.3390/agronomy12020515_

Round 1

Reviewer 1 Report

Although the idea of this work is interesting and this study may be appropiate for agronomy, this work has relevant flaws that must be addressed before it can be considered for publication. The authors can find a list with major concerns that must be solved.

  1. To establish a correlation between phenolics and browning, it is necessary to add at least 10 different conditions in the study. Problably, the use of other cultivars can be beneficial. The number of data is not enough to support the conclusions. Further, the data was not processed by regression analysis.
  2. The manuscript has a lot of English laguange problems that must be solved. The organization of the manuscript is not completely right. For example, the sentences and topics are not connected in the introduction. "Thus" is inappropriately used along the manuscript. Several sentences need grammar revision.
  3. The abstract is not well written. It does not show the work that have been done in this study. The sentences regarding the introduction to the research field are not clear.
  4. To try to correlate the phenolic compounds with the browning seem not enough to complete the study. SOD, CAT, H2O2, ROS contents, among other possible experiments, must be also added to provide enough results to confirm the conclusions. A HPLC-based analysis of the specific phenolics found in artichoke extracts would be interesting.
  5. I don't understand what the authors are trying to show in Table 2. In addition, there is no deviation, no units...
  6. The keywords are not well chosen. It is quite weird that postharvest, preservation or artichoke do not appear among the chosen keywords.
  7. It is not clear how many times the experiments were repeated.
  8. There are numerous reports in this field that are missed in the manuscript. For example, DOI numbers: 10.1016/j.foodchem.2013.04.028, 10.1016/j.postharvbio.2012.07.006, 10.1002/jsfa.9775 and 10.1016/j.postharvbio.2014.08.008.

Author Response

REVIEWER 1

Dear reviewer,

Thank you very much for your useful comments which have aid to improve our original manuscript. Below you can find an itemed list of your comments and suggestions and the answer and modification performed in the revised manuscript according to your suggestions. The new information added to the revised manuscript is highlighted in blue ink.

Although the idea of this work is interesting and this study may be appropriate for agronomy, this work has relevant flaws that must be addressed before it can be considered for publication. The authors can find a list with major concerns that must be solved.

  1. To establish a correlation between phenolics and browning, it is necessary to add at least 10 different conditions in the study. Probably, the use of other cultivars can be beneficial. The number of data is not enough to support the conclusions. Further, the data was not processed by regression analysis.

Response: A correlation analysis was carried out between phenolic content and increase on hue angle for main, secondary and tertiary artichoke heads. In addition, because new analytical determinations have been made (PPO enzyme activity), another correlation analysis has been carried out between phenolic content and PPO activity for main, secondary and tertiary artichoke heads.

These correlations have been included in section 2.5. Statistical analysis and the results (Line 176-178), and the results obtained have been included in section 3.3. (Line 300-302) and 3.4 (Line 367-372).

  1. The manuscript has a lot of English language problems that must be solved. The organization of the manuscript is not completely right. For example, the sentences and topics are not connected in the introduction. "Thus" is inappropriately used along the manuscript. Several sentences need grammar revision.

Response:  The whole manuscript has been revised for English grammar and style corrections by seniors, and changes have been highlighted in red ink.

  1. The abstract is not well written. It does not show the work that have been done in this study. The sentences regarding the introduction to the research field are not clear.

 Response: The abstract has been reviewed and the introduction has been updated at research field and more details have been added to improve the experimental design of this study.  

  1. To try to correlate the phenolic compounds with the browning seem not enough to complete the study. SOD, CAT, H2O2, ROS contents, among other possible experiments, must be also added to provide enough results to confirm the conclusions. A HPLC-based analysis of the specific phenolics found in artichoke extracts would be interesting.

Response: To try to correlate phenolic compounds with browning, new analyzes have been carried out to determine the main enzyme involved in browning. Polyphenol oxidase (PPO) activity has been analysed for main, secondary and tertiary heads. A new section was added in Materials and Method: 2.4. PPO activity (Line: 156-166). After the analysis of PPO activity, the statistical analysis and discussion of these results have been included in the manuscript in a new section 3.3. PPO activity for different artichoke head orders (Line: 283-305).

On the other hand, a HPLC-based analysis of phenolic compounds has not been done due to hydroxycinnamic acids and luteolin derivatives were quantified by RP-HPLC-DAD in 8 cultivars obtained by different propagation methods in a previous work including ‘Lorca’ cultivar phenolic profile (Giménez et al., 2021). Some information has been added in the introduction related to the results published in the previous work (Line 79-82).

Giménez, M.J., Giménez-Berenguer, M., García-Pastor, M.E., Parra, J., Zapata, P.J., Castillo, S. The influence of flower head order and gibberellic acid treatment on the hydroxycinnamic acid and luteolin derivatives content in globe artichoke cultivars. Foods 2021, 10(8), 1813. doi:10.3390/foods10081813

  1. I don't understand what the authors are trying to show in Table 2. In addition, there is no deviation, no units...

Response: More details of the analysis of variance carried out have been included in section 2.5. statistical analysis (Line 169-170; 172).

In addition, some information has been added in the section 3.1. related to results and discussion and units were specified both in text and table 2 (Line 182-183).

  1. The keywords are not well chosen. It is quite weird that postharvest, preservation or artichoke do not appear among the chosen keywords.

Response: The keywords have been reviewed and some of the proposed word have been included.

Keywords: total phenol content; browning susceptibility; artichoke head order; internal development stage; postharvest

  1. It is not clear how many times the experiments were repeated.

Response: In the manuscript it has been specified that the experiment was carried out throughout the de developmental cycle of one season (Line 92). All artichokes of each plant were harvested and analyzed as they reached commercial size.

  1. There are numerous reports in this field that are missed in the manuscript.

For example, DOI numbers:

Bach, V.; Jensen, S.; Clausen, M.R.; Bertram, H.C.; Edelenbos, M. Enzymatic browning and after-cooking darkening of Jerusalem artichoke tubers (Helianthus tuberosus L.) Food Chemistry 2013, 141, 1445.      doi:10.1016/j.foodchem.2013.04.028

Cabezas-Serrano, A.B.; Amodio, M.L.; Colelli, G. Effect of solution pH of cysteine-based pre-treatments to prevent browning of fresh-cut artichokes. Posharvest Biology and Technology 2013, 17-23. doi:10.1016/j.postharvbio.2012.07.006

Rizzo, V.; Lombardo, S.; Pandino, G.; Barbagallo, R. N.; Mazzaglia, A.; Restuccia, C.; Mauromicale, G.; Muratore, G. Shelf-life study of ready-to-cook slices of globe artichoke ‘Spinoso sardo’: effects of anti-browning solutions and edible coating enriched with Foeniculum vulgare essential oil. Science of Food and Agriculture 2019, 99, 5219-5228. doi:10.1002/jsfa.9775

Ghidelli, C.; Mateos, M.; Rojas-Argudo, C.; Pérez-Gago, M.B. Novel approaches to control browning of fresh-cut artichoke: Effect of a soy protein-based coating and modified atmosphere packaging. Postharvest Biology and Technology 2015, 99, 105-113. doi:10.1016/j.postharvbio.2014.08.008.

Response: All references proposed have been included in the manuscript except Bach et al. 2013 which refers to another crop (Jerusalem artichoke). In addition, new references have been included in the manuscript to improve the discussion of the results (References 34: Line 503-504; References 42-50: Line 519-541).

Reviewer 2 Report

I think this is a well written paper containing original results. 

Author Response

Dear reviewer,

Thank you very much for your useful comments which could lead to publish our recently obtained results and thanks for your compliments.

I think this is a well written paper containing original results.

Reviewer 3 Report

This is an interesting and well written study. Nevertheless, there is a lack of novelty and the data collection could be improved. The objective of this study is very interesting and the agronomical experiment design is appropriate. However, to measure only TPC is inadequate for high-quality research. TPC alone is not enough for a characterization, the authors should perform other analysis such as phenolic compounds (hydroxycinnamic acids and flavonoids) profile by LC and/or PPO o POD enzymatic activities. In addition, the correlation between the TPC in flower head order and the browning was not statistical.

Author Response

REVIEWER 3

Thank you very much for your useful comments which have aid to improve our original manuscript. Below you can find an itemed list of your comments and suggestions and the answer and modification performed in the revised manuscript according to your suggestions. The new information added to the revised manuscript is highlighted in blue ink since the changes are similar as the suggested by reviewer 1.

This is an interesting and well written study. Nevertheless, there is a lack of novelty and the data collection could be improved. The objective of this study is very interesting and the agronomical experiment design is appropriate.

 However, to measure only TPC is inadequate for high-quality research. TPC alone is not enough for a characterization, the authors should perform other analysis such as phenolic compounds (hydroxycinnamic acids and flavonoids) profile by LC and/or PPO o POD enzymatic activities.

Response: A HPLC-based analysis of phenolic compounds has not been done due to hydroxycinnamic acids and luteolin derivatives were quantified by RP-HPLC-DAD in 8 cultivars obtained by different propagation methods in a previous work including ‘Lorca’ cultivar phenolic profile (Giménez et al., 2021). Some information has been added in the introduction related to the results published in the previous work (Line 79-82).

Giménez, M.J., Giménez-Berenguer, M., García-Pastor, M.E., Parra, J., Zapata, P.J., Castillo, S. The influence of flower head order and gibberellic acid treatment on the hydroxycinnamic acid and luteolin derivatives content in globe artichoke cultivars. Foods 2021, 10(8), 1813. doi:10.3390/foods10081813

Response: To try to correlate phenolic compounds with browning, new analyzes have been carried out to determine the main enzyme involved in browning. Polyphenol oxidase (PPO) activity has been analysed for main, secondary and tertiary heads. A new section was added in Materials and Method: 2.4. PPO activity (Line: 156-166). After the analysis of PPO activity, the statistical analysis and discussion of these results have been included in the manuscript in a new section 3.3. PPO activity for different artichoke head orders (Line: 283-305).

 In addition, the correlation between the TPC in flower head order and the browning was not statistical.

Response: A correlation analysis was carried out between phenolic content and increase on hue angle for main, secondary and tertiary artichoke heads. In addition, because new analytical determinations have been made (PPO enzyme activity), another correlation analysis has been carried out between phenolic content and PPO activity for main, secondary and tertiary artichoke heads.

These correlations have been included in section 2.5. Statistical analysis and the results (Line 176-178), and the results obtained have been included in section 3.3. (Line 300-302) and 3.4 (Line 367-372).

Round 2

Reviewer 1 Report

The authors have addressed all my comments. I think the manuscript can be published in the current state.